# Human cytomegalovirus-IE2 suppresses antigen presentation of macrophage through the IL10/STAT3 signalling pathway in transgenic mouse

Xianjuan Zhang[1], Qing Wang[1], Shuo Han[2], Guanghui Song[1], Bin Wang[3]*, Yunyang Wang[4]*

1 Department of Clinical Laboratory, The Affiliated Hospital of Qingdao University, Qingdao, China,
2 Department of Spinal Surgery, The Affiliated Hospital of Qingdao University, Qingdao, China,
3 Department of Pathogenic Biology, Department of Special Medicine, School of Basic Medicine, Qingdao University, Qingdao, China, 4 Department of Endocrinology and Metabolism, The Affiliated Hospital of Qingdao University, Qingdao, China

* wangbin532@126.com (BW); wangyy_09@outlook.com (YW)

## Abstract

Human cytomegalovirus (HCMV) has evolved sophisticated strategies to evade host immune defenses, enabling its persistent survival in human populations. HCMV intermediate-early protein 2 (IE2) has been identified as a crucial factor in immune evasion mechanisms. However, the specific immunomodulatory effects of IE2 on antigen presentation remain insufficiently explored. In this study, we established a transgenic mouse model to systematically examine the impact and molecular mechanisms of IE2 on macrophages (Mφs) antigen presentation *in vivo*. Our findings demonstrated that IE2 modifies Mφs' function by preventing their phagocytic activity and polarization. Additionally, IE2 inhibits Mφs overactivation both *in vivo* and *in vitro*, which raises IL-10 levels and activates the downstream mediator STAT3, which in turn decreases T cell immune responses by encouraging T helper 2 (Th2) type responses. In conclusion, these findings underscore the potential of IE2 as a critical regulator of immune evasion and may contribute to the development of novel, targeted therapeutic strategies against the virus.

## Introduction

Human cytomegalovirus (HCMV), a β-herpesvirus, represents a significant opportunistic pathogen that can cause life-threatening complications in immunocompromised individuals, including organ and stem cell transplant recipients and patients with acquired immunodeficiency syndrome [1–3]. HCMV encodes more than 250 open reading frames (ORFs), most of which are involved in regulating virus-host interactions, evading host immune responses, and facilitating viral replication and dissemination [4–7].

**Data availability statement:** All relevant data are within the paper and its Supporting Information files.

**Funding:** This research was funded by Shandong Provincial Science and technology Foundation (grant no. 2019JZZY011009), Qingdao Municipal Science and technology Foundation (grant no. 20-2-3-4-nsh), National Key Research and Development Program of China (grant no.2018YFA0900802), Shandong Provincial Natural Science Foundation (grant no. ZR2021QH254).

**Competing interests:** The authors have declared that no competing interests exist.

The intermediate-early protein 2 (IE2), a 579-amino acid protein encoded by the HCMV-*UL122* gene, plays an indispensable role in viral replication. Its unique functions cannot be compensated by any other viral or cellular proteins [8]. Previous research has demonstrated that IE2 facilitates immune evasion through STING degradation and subsequent inhibition of cGAMP-mediated IFN-β induction [9]. However, the precise mechanisms underlying IE2-mediated immune modulation remain poorly understood.

Macrophages (Mφs) have emerged as crucial players in HCMV persistence and dissemination [10]. As professional antigen-presenting cells (APCs), Mφs not only initiate immune responses but also serve as critical mediators between innate and adaptive immunity [11,12]. HCMV has evolved sophisticated mechanisms to manipulate Mφ function, including the prolonged expression of Mcl-1 through EGFR activation, which disrupts normal apoptotic pathways and creates long-lived macrophages that serve as persistent viral reservoirs [13–15]. Therefore, the function of Mφs is crucial to the HCMV-induced immune evasion strategies.

Given the strict species-specific tropism of the virus, an HCMV-infected animal model has not yet been developed, leaving the *in vivo* mechanisms underlying HCMV-induced immune evasion unclear. Cytomegaloviruses (CMVs) of different species differ in their virulence, illness processes, and pathology, using animal CMV-infected models provide some insights, substantial differences in virulence, disease progression, and pathology limit their utility in understanding HCMV-specific mechanisms [16]. Hence, a C57BL/6-Tg (HCMV-*UL122*) mouse model (termed IE2 mice) that stably expresses the IE2 protein successfully constructed in our laboratory. Based on this model, we examine the effects of persistent IE2 expression on immune activation and to investigate other potential mechanisms *in vivo*.

In this study, we demonstrate long-term IE2 expression significantly impairs immune activation. Our findings reveal that IE2 inhibits macrophage antigen presentation through activation of the IL-10/STAT3 signaling pathway, consequently suppressing T-cell responses. These results provide novel insights into the mechanisms of HCMV immune evasion and highlight the critical role of IE2 in modulating host immune responses.

## Materials and methods

### Animals

This study was carried out in strict accordance with the recommendations in the Guide for the Care and Use of Laboratory Animals of the National Institutes of Health. The protocol was approved by the Committee on the Ethics of Animal Experiments of Qingdao University (Protocol Number: 20220928C5716820230223345119). Mice were housed under controlled conditions with a 12-hour light/dark cycle, ad libitum access to food and water, and daily monitoring for signs of distress or illness. At the end of the experimental period, euthanasia was performed to minimize suffering. Animals were first anesthetized with sodium pentobarbital (50 mg/kg, intraperitoneal injection) to ensure unconsciousness and pain relief, followed by cervical dislocation as a secondary method to confirm

death. This procedure adheres to the guidelines of the American Veterinary Medical Association (AVMA) and was carried out by trained personnel to ensure humane treatment.

## Construction of IE2 transgenic mice

The IE2 gene was derived from human herpesvirus 5 strain, GenBank accession: NC_006273.2. The pAV.Ex1d-CMV-IE2 vector containing the IE2 gene and the expression vector pAV.Des1d were constructed. Both vectors were digested with BamH I, and the digested products were separated by 0.8% agarose gel electrophoresis. The target fragments were excised and purified separately. The target fragments and vector fragments were ligated at a molar ratio of 3:1. The ligation products were transformed into competent E. coli DH5α cells, and single colonies were selected for culture expansion. Recombinant plasmids were extracted using a plasmid DNA extraction kit (QIAGEN, Cat no.12123). Polymerase chain reaction (PCR) amplification was performed using primers (5′-CACTTTGTA CAAGAAAGCTGG-3′ and 5′-CAAGTTTGTACAAAAAAGCAGGCT-3′) to screen for plasmids containing the target fragment. The selected plasmids were subjected to sequencing analysis, and those with the correct sequence were used for the construction of the transgenic mouse model. The plasmid was injected into fertilized eggs of C57/BL6 mice using prokaryotic microinjection, and embryos were transferred into C57BL/6 mice to obtain the F0 generation C57BL/6-Tg (HCMV-*UL122*) transgenic mice (IE2 mice). To obtain the F1 generation, F0 mice were mated with wild-type (WT) mice (S1 Fig). IE2-positive mice were genotyped by polymerase chain reaction PCR. The PCR primers used were 5′-GCAATTCTTTGAGGCTCCAC-3′ and 5′-CCGCAAGAACAAGAGCAAAC-3′ (IE2 PCR product length: 470 bp).

## Extraction and culture of primary macrophages

Primary peritoneal Mφs were isolated from six- to eight-week-old female C57BL/6 mice. Mice were intraperitoneally injected with 1 mL of thioglycolate (100 mg/mL), and peritoneal exudate was collected 7 days later. The primary peritoneal Mφs were resuspended in Dulbecco's modified Eagle's medium (DMEM) with 10% fetal bovine serum (FBS, Gibco by Invitrogen, Carlsbad, CA).

## Immunisation

For *in vitro* experiments, primary peritoneal Mφs were cultured in six-well plates at a density of $5 \times 10^5$ cells/mL and incubated overnight. Primary cells were treated with 2 μg/mL of lipopolysaccharide (LPS) for 48 h and divided into WT+LPS and IE2+LPS groups. Phosphate-buffered saline (PBS) was used as a blank control and divided into WT+PBS and IE2+PBS groups.

For *in vivo* experiments, mouse splenic lymphocytes were isolated 12 h following LPS (0.01 mg/g) immunisation and passed through a 70-μm cell strainer to prepare a single-cell suspension. Six- to eight-week-old female IE2 mice were randomly divided into two groups: IE2 and IE2+T (n = 5 in each group). WT C57BL/6 mice of the same age were randomly divided into two groups: WT and WT+T (n = 5 in each group). The IE2 and WT groups were injected with PBS, while the IE2+T and WT+T groups were injected with LPS.

## Enzyme-linked immunosorbent assay (ELISA) for cytokine production

Primary peritoneal Mφs were treated with 2 μg/mL of LPS or PBS for 48 h *in vitro*. The levels of TNF-α (Abcam, Cat no. ab100747), IFN-γ (Thermo Fisher Scientific, Cat no. KMC4021C), IL-6 (Abcam, Cat no. ab222503) and IL-12p70 (Abcam, Cat no. ab119531) expression were measured using their respective enzyme-linked immunosorbent assay (ELISA) kits.

## Cell staining and flow cytometry

Single-cell suspensions were seeded into 96-well plates at a density of $1 \times 10^5$ cells/well in 100 μL of 10% foetal bovine serum-containing DMEM. Extracellular staining was performed using the following antibodies: anti-F4/80-PE (BioLegend; Cat no. 123109), anti-CD206-APC (BioLegend; Cat no. 141707), anti-CD11b-FITC (BioLegend; Cat no. 101206),

anti-CD80-PE (BioLegend; Cat no. 104708), anti-CD86-PE (BioLegend; Cat no. 105008), anti-H-2Kb-PE (BioLegend; Cat no. 116508), anti-I-A/I-E-PE (BioLegend; Cat no. 107608), anti-CD3-FITC (BioLegend; Cat no. 100203), anti-CD4-APC (BioLegend; Cat no. 100411), and anti-CD8-APC (BioLegend; Cat no. 100714).

Intracellular staining was performed using the following antibodies: anti-NOS2 (iNOS)-APC (BioLegend; Cat no. 696807), anti-IL-4-PE (BioLegend; Cat no. 504103), anti-IFN-γ-PE (BioLegend; Cat no. 505808), and anti-TNF-α-PE (BioLegend; Cat no. 506305). All antibodies were anti-mouse and used according to the manufacturer's instructions. The fluorescence was measured using a Beckman CytoFLEX flow cytometer, and data were analysed using the FlowJo (v10) software. The gating strategy is illustrated in S2 Fig.

### Transwell migration assay

For the MCP-1-dependent migration assay of primary Mφs by transwell (8.0 μm, Corning, ME, USA). $2 \times 10^5$ Mφs were seeded in the inserts with 100 μL of serum-free DMEM media. The lower chamber contained 600 μL DMEM media with 10% FBS, and 20 ng/mL MCP-1 (Gibco, Invitrogen, USA) was added. After 20 min, the migratory cells in the bottom chamber were counted.

### Macrophage phagocytic function

*Escherichia coli* (*E. coli*) was labelled with fluorescein isothiocyanate (FITC; Yeasen) [17]. The density of *E. coli* in PBS was adjusted to $1 \times 10^8$ cells/mL. The bacteria were then inactivated at 60°C for 15 min, and 1 mL of the suspension was mixed with 100 μL of 1 mg/mL FITC in DSMO and incubated in the dark for 1 h at $18-22$°C. After three washes with PBS and centrifugation at $6,000 \times g$ for 5 min, the FITC-labelled bacteria were resuspended in 1 mL of PBS and stored at −20°C. Subsequently, 50 μL of the FITC-labelled *E. coli* suspension ($1 \times 10^8$ cells/mL) was seeded into a Mφ-containing culture dish ($1 \times 10^6$ cells/well) for infection. The mixture was incubated in the dark for 1 h at room temperature and shaken every 5 min to prevent the settling of the Mφs and *E. coli*. After 30 min, the cells were washed with 300 μL of PBS, and 1 μL of cold trypan blue (0.4% in PBS) was added to each sample to quench the fluorescence from extracellular bacteria. A fluorescence microscope was used to observe the fluorescence. Phagocytic capacity was defined as the percentage of Mφs that phagocytosed one or more bacteria in the total cell population.

### Real-time quantitative PCR (RT-qPCR)

RNA was extracted using the RNeasy kit (TIANGEN), and cDNA was synthesised using the SuperScript kit (Roche). Gene expression was quantified by RT-qPCR on a Roche 480 Real-Time System. Primers were synthesised by Sangon Biotech. The $2^{-\Delta\Delta CT}$ method was used to determine the relative expression of target genes, normalised to GAPDH. The primers used are listed in Table 1.

### Western blot analysis

Protein was extracted in cells were determined by western blot analysis. Briefly, proteins were subjected to sodium dodecyl sulfate–polyacrylamide gel electrophoresis and electroblotted to nitrocellulose membranes. Primary antibodies: anti-Rabbit STAT3 (1:1000, ABclonal, Wuhan, China), anti-Rabbit p-STAT3-Y705 (1:1000, ABclonal, Wuhan, China), anti-Mouse GAPDH (1:10000, ABclonal, Wuhan, China), anti-Rabbit β-actin (1:10000, ABclonal, Wuhan, China), and anti-Mouse CMV-IE1 and IE2 (1:100, cat. ab53495; Abcam, Cambridge, UK). Horseradish-peroxidase–conjugated anti-rabbit IgG and anti-mouse IgG (Beyotime, Shanghai, China) were used as the secondary antibody. Immune complexes were detected using a western chemiluminescent horseradish-peroxidase substrate (Millipore Corporation, Billerica, MA). Densitometry of immunoblot analysis was performed using ImageJ software (National Institutes of Health, Bethesda, MD).

**Table 1. The sequence of the primers for RT-qPCR.**

| Gene name | Primers (5'-3') |
|---|---|
| β-2m | CCTGGTCTTTCTGGTGCTTGTCTC (Forward) <br> CAATGTGAGGCGGGTGGAACTG (Reverse) |
| H2-D1 | CTTGCGATATGCTGTGGTGCTTA (Forward) <br> ATGAAGAATGTGGTGAAGTTGATGTGC (Reverse) |
| CD74 | TGCTGATGCGTCCAATGTCCATG (Forward) |
| | GGCTGACTTCTTCCTGGCACTTG (Reverse) |
| IL10 | CTGGACAACATACTGCTAACCGACTC (Forward) <br> ACTGGATCATTTCCGATAAGGCTTGG (Reverse) |
| IL12b | GACCATCACTGTCAAAGAGTTTCTAGAT (Forward) <br> AGGAAAGTCTTGTTTTTGAAATTTTTTAA (Reverse) |
| STAT3 | CGTAGTGACAGAGAAGCAGCAGATG (Forward) <br> TGAAATCAAAGTCGTCCTGGAGGTTC (Reverse) |
| Cfap47 | CCTGACGACTCAATTCCATATACTCTC (Forward) <br> GACTCTTCCTCATCTCTGCTCTTAAC(Reverse) |
| Fam47c | CCCGCCAAGCAGTCTCTTTATTAC (Forward) <br> GAAATGTTCATCAATGCCCAAGTCTC(Reverse) |
| GAPDH | GGTTGTCTCCTGCGACTTCA (Forward) <br> TGGTCCAGGGTTTCTTACTCC (Reverse) |

## Statistical analysis

Data are expressed as means ± SEM and analysed using one-way analysis of variance. A paired t-test was used for comparative analysis between two groups. Non-normally distributed data were analysed using nonparametric tests. All statistical analyses were performed using the GraphPad Prism (v7.0) software. A $P$-value < 0.05 was considered statistically significant.

## Results

### Evaluation of Mφ activation *in vivo*

To investigate the role of IE2 in immune activation *in vivo*, we developed a C57BL/6-Tg (HCMV-*UL122*) mouse model (IE2 mice) that stably expresses the IE2 protein. Whole-genome sequencing using the Illumina NovaSeq6000 platform revealed that the UL122 gene vector was inserted into the X chromosome, located 173,437 bp downstream of the *Fam47c* gene and 344,827 bp upstream of the *Cfap47* gene (Fig 1A and 1B). There was no difference in the mRNA levels of *Fam47c* and *Cfap47* between WT and IE2 mice (Fig 1C). These results indicated that the *UL122* gene insertion does not affect the expression of neighbouring genes. In addition, we did not observe significant changes in weight or dietary intake between IE2 mice and WT mice of the same age (S3 Fig). Genotyping by PCR confirmed the presence of the IE2 transgene (Fig 1D), and IE2 protein expression in Mφs was validated by western blotting (Fig 1E).

Next, the mice were immunised with LPS *via* intraperitoneal injection. After 12 h, we examined the expression of surface markers, including CD80, CD86, MHC I, and MHC II, in CD11b⁺ splenocytes using flow cytometry (Fig 2A). CD11b was utilized as markers for macrophages [18]. The IE2 + T group exhibited a significant reduction in the number of CD11b⁺ cells compared to the WT + T group. Furthermore, MHC I and MHC II expression was markedly downregulated in CD11b⁺ cells from the IE2 + T group, while no significant differences were observed in CD80 and CD86 expression (Fig 2B). These results suggest that IE2 inhibits Mφ activation and downregulates MHC I and MHC II expression *in vivo*.

Mφs release pro-inflammatory cytokines to enhance immune cell function and antigen capture. To evaluate the effect of IE2 on cytokine secretion, primary peritoneal Mφs were cultured with LPS for 48 hours *in vitro*, and cytokine levels in

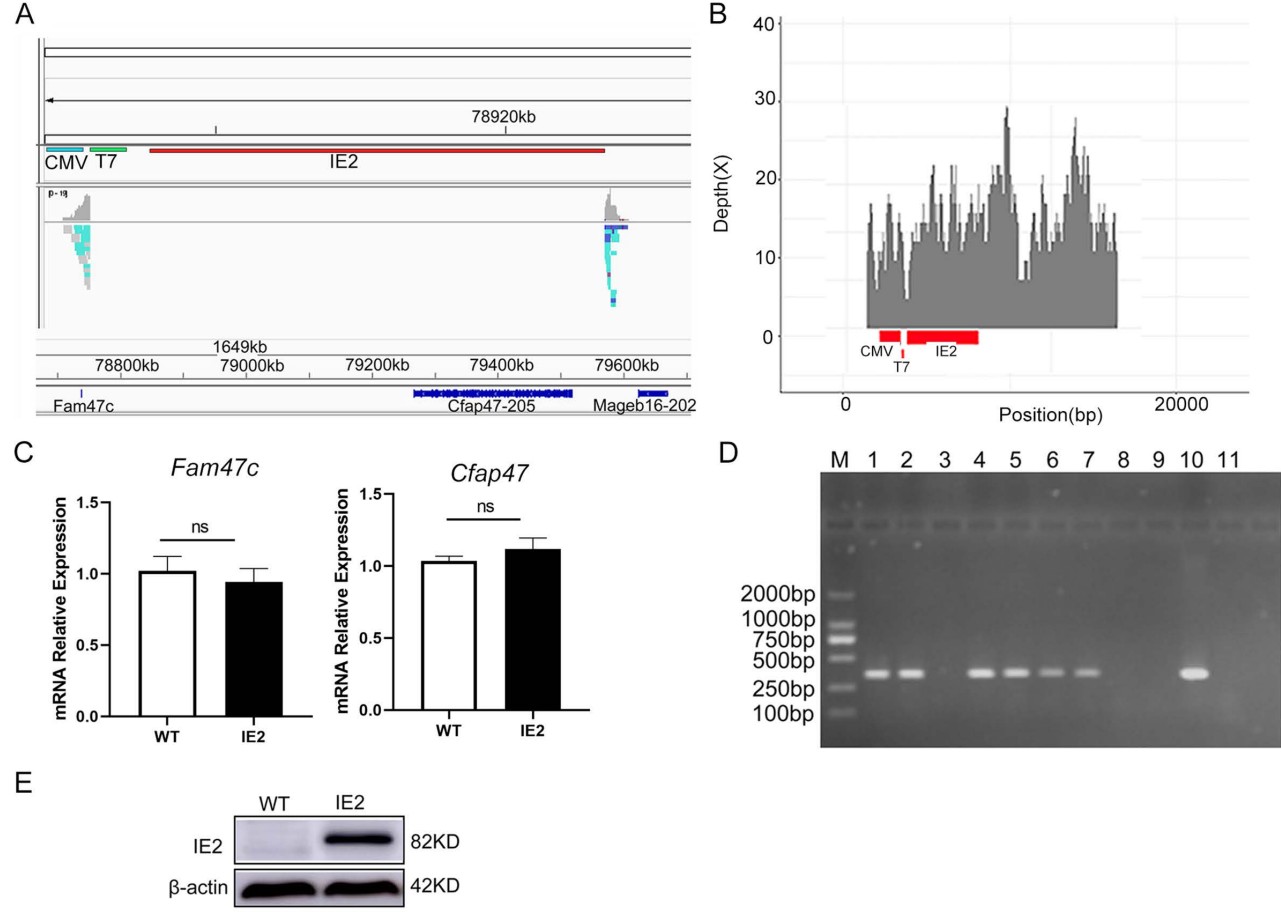

**Fig 1. Identification of C57BL/6-Tg (HCMV-UL122) mice.** (A) Illumina NovaSeq6000 sequencing platform to sequence the whole genome of the F0 generation transgenic mice. (B) Detection of transgenic fragments. The fragment's detection is indicated by intervals with sequencing depth. (C) The expression of *Fam47c* and *Cfap47* mRNA level was detected by RT-qPCR. Date are mean±SEM. (D) Identification of the IE2 positive and negative mice by PCR. (Lanes 1-2, 4-7, 10) positive mice; (Lanes 3, 8) the negative mice; (Lanes 10, 11) the positive and water control. PCR product size: 470 bp. (E) IE2 expressed in macrophages were confirmed by western-blot technology.

the supernatants were measured by ELISA. Compared to the WT+LPS group, the IE2+LPS group exhibited significantly reduced secretion of TNF-α, IFN-γ, IL-6, and IL-12p70 (Fig 2C). These findings indicate that IE2 suppresses the secretion of multiple pro-inflammatory cytokines.

## Effect of IE2 on function of Mφ

To assess the effect of IE2 on Mφ polarization *in vivo*, the proportions of iNOS+ (M1) and CD206+ (M2) cells within the F4/80+ population in the spleen were analyzed. The IE2+T group showed a significant decrease in M1 phenotype Mφs and an increase in M2 phenotype Mφs compared to the WT+T group (Fig 3A,B), suggesting that IE2 promotes polarization toward the M2 phenotype while inhibiting the M1 phenotype.

Next, we evaluated the phagocytic and migratory capacities of primary peritoneal Mφs. Phagocytic function was assessed using FITC-labeled *E. coli*, and migration was measured using a transwell assay with MCP-1 as a chemoattractant. Fluorescence microscopy revealed a significant reduction in phagocytic activity in IE2 mice compared to WT mice (Fig 3C,D). Conversely, the number of migrating Mφs in response to MCP-1 was significantly higher in the IE2+MCP-1

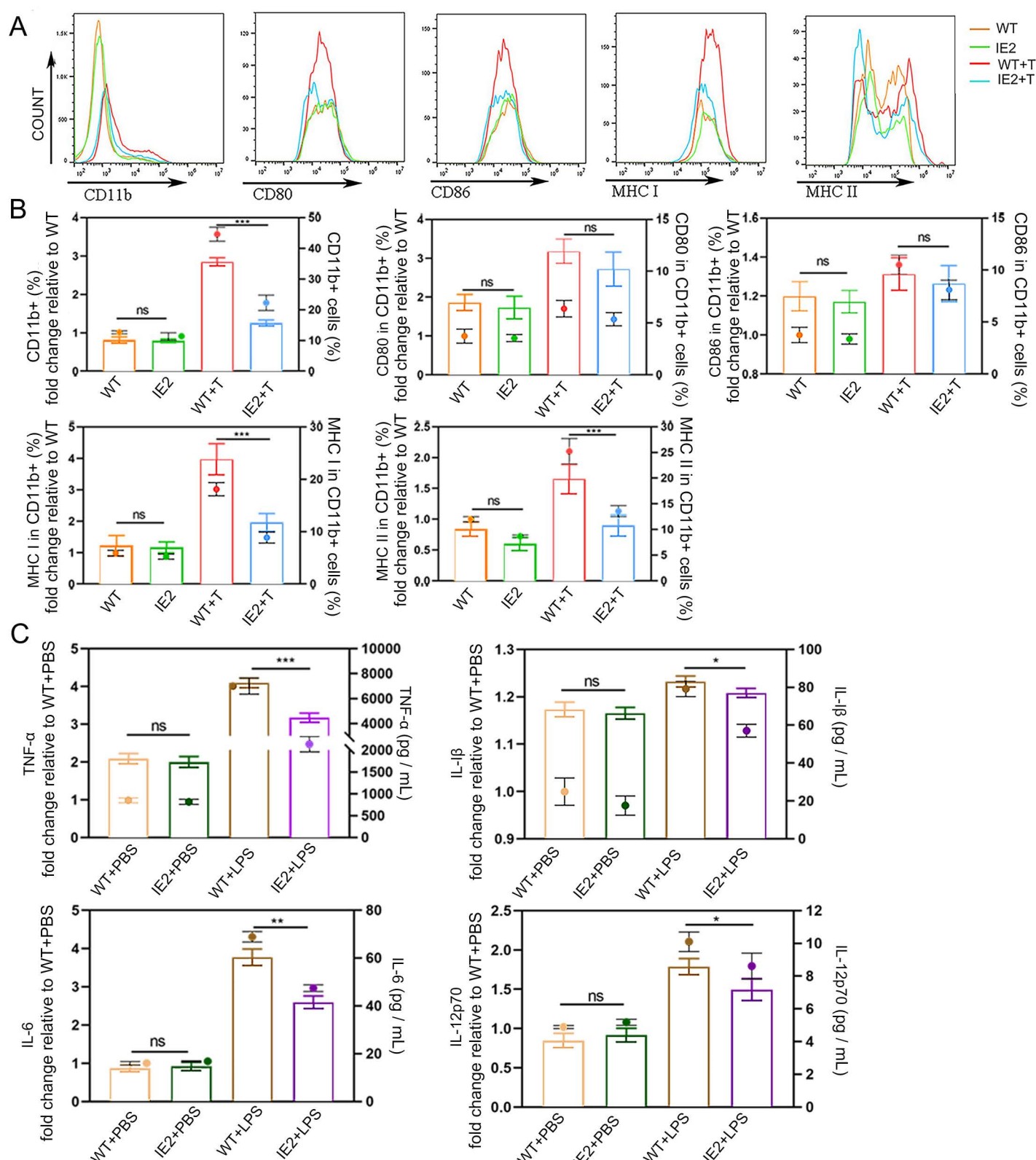

**Fig 2. IE2 inhibits the activation of macrophages *in vivo*.** (A) All mice were immunized with LPS *via* intraperitoneal injection, after 12h, spleen lymphocytes were isolated, and the surface markers CD80, CD86, MHC I, and MHC II in CD11b⁺ macrophages were detected by flow cytometry. (B) The

percentages of CD80, CD86, MHC I and MHC II expression in CD11b+ macrophages. WT + T means WT mice injected intraperitoneally with LPS. IE2 + T means IE2 mice injected intraperitoneally with LPS. (C) Assessment of TFN-α, IL-1β, IL-6 and IL-12p70 were detected in primary peritoneal macrophages after LPS-treated 48h *in vitro*. Statistically significant differences as determined by t-test from ANOVA. Date are mean ± SEM. n = 5. Significance levels were defined as *$P < 0.05$, **$P < 0.01$ and ***$P < 0.001$.

group than in the WT + MCP-1 group (Fig 3E,F), indicating that IE2 enhances Mφ migratory capacity while impairing phagocytic function.

### Effect of IE2 on CD4+ and CD8+ T cells

Given the observed downregulation of MHC I and MHC II expression in Mφs, we investigated the impact of IE2 on T cell responses. CD4+ T cells play a crucial role in controlling CMV infection by recognizing viral protein-derived antigens presented via MHC II molecules on APCs. In contrast, CD8+ T cells are stimulated to differentiate into cytotoxic T cells for killing infected cells through the recognition of MHC I molecules on APCs. Activation of Th1 cells can also enhance the killing function of CD8 T cells, whereas Th2 cells inhibit the activity of macrophages and Th1 cells. Primary peritoneal Mφs were stimulated with 3 μg/mL LPS for 48 hours *in vitro*. Then, $2 \times 10^5$ of these stimulated Mφs were co-cultured with $2 \times 10^6$ autologous T cells (at a 1:10 ratio) isolated from the spleens of mice. The cells were harvested after 48 hours for flow cytometry analysis. The results revealed that, compared to the WT + LPS group, the IE2 + LPS group exhibited significantly reduced expression of the Th1 cytokine IFN-γ and increased expression of the Th2 cytokine IL-4 in CD4+ T cells (Fig 4A,B). Additionally, the expression of IFN-γ and TNF-α in CD8+ T cells was lower in the IE2 + LPS group (Fig 4C,D), suggesting that IE2 suppresses cytotoxic T cell activity. These results indicate that IE2 promotes Th2-type immune responses while inhibiting Th1-type responses, thereby impairing T cell-mediated immunity.

### Inhibition of T cell responses *via* up-regulation of the IL-10/STAT3 signalling pathway

The results thus far indicate that IE2 inhibits antigen presentation of Mφs *in vivo*. To elucidate the mechanism underlying the downregulation of MHC I and MHC II expression, we analyzed the mRNA levels of *β-2m*, *H2-D1*, *CD74*, *IL-12b*, and *IL-10*. The IE2 + LPS group exhibited reduced expression of *β-2m*, *CD74*, and *IL-12b* but significantly increased expression of *IL-10* compared to the WT + LPS group (Fig 5A,B). Furthermore, IL-10, a potent anti-inflammatory cytokine, activates its primary mediator, STAT3, to inhibit an activated immune response [19]. We detected the Y705 phosphorylated form of STAT3 (p-STAT3) and total STAT3 expression in T cells. We found that the expression of p-STAT3 was significantly increased in the IE2 + LPS group (Fig 5C,D). These findings suggest that IE2 may inhibit Mφs and T cells *via* the IL-10/STAT3 signalling pathway.

## Discussion

During HCMV infection, Mφs are believed to be crucial for viral persistence and dissemination [10]. As the first line of defense, Mφs are key modulators of antiviral immune responses, balancing protection and viral pathogenesis [20]. They phagocytose cellular debris and pathogens, process and present antigens to T cells, and activate effector immune cells to mount a robust response against infections [11]. To explore the specific role of IE2 in host immunity *in vivo*, we established IE2 mice as an animal model. In this study, we investigated the effects of the IE2 protein of HCMV on Mφ function. Compared with WT mice, the antigen presentation and phagocytic function of Mφs were markedly reduced in IE2 mice. Furthermore, IE2 skewed Mφ polarization toward the M2 phenotype while suppressing the M1 phenotype. IL-10 levels were elevated in IE2 mice, leading to increased p-STAT3 expression. These findings suggest that IE2 impairs Mφ function in IE2 mice through IL-10 and p-STAT3 upregulation, resulting in a diminished host immune response.

The establishment of lifelong HCMV persistence relies on a delicate balance between host immune defenses and viral immune evasion mechanisms [6]. Innate immune cells, including DCs, Mφs, and natural killer cells, are highly efficient at

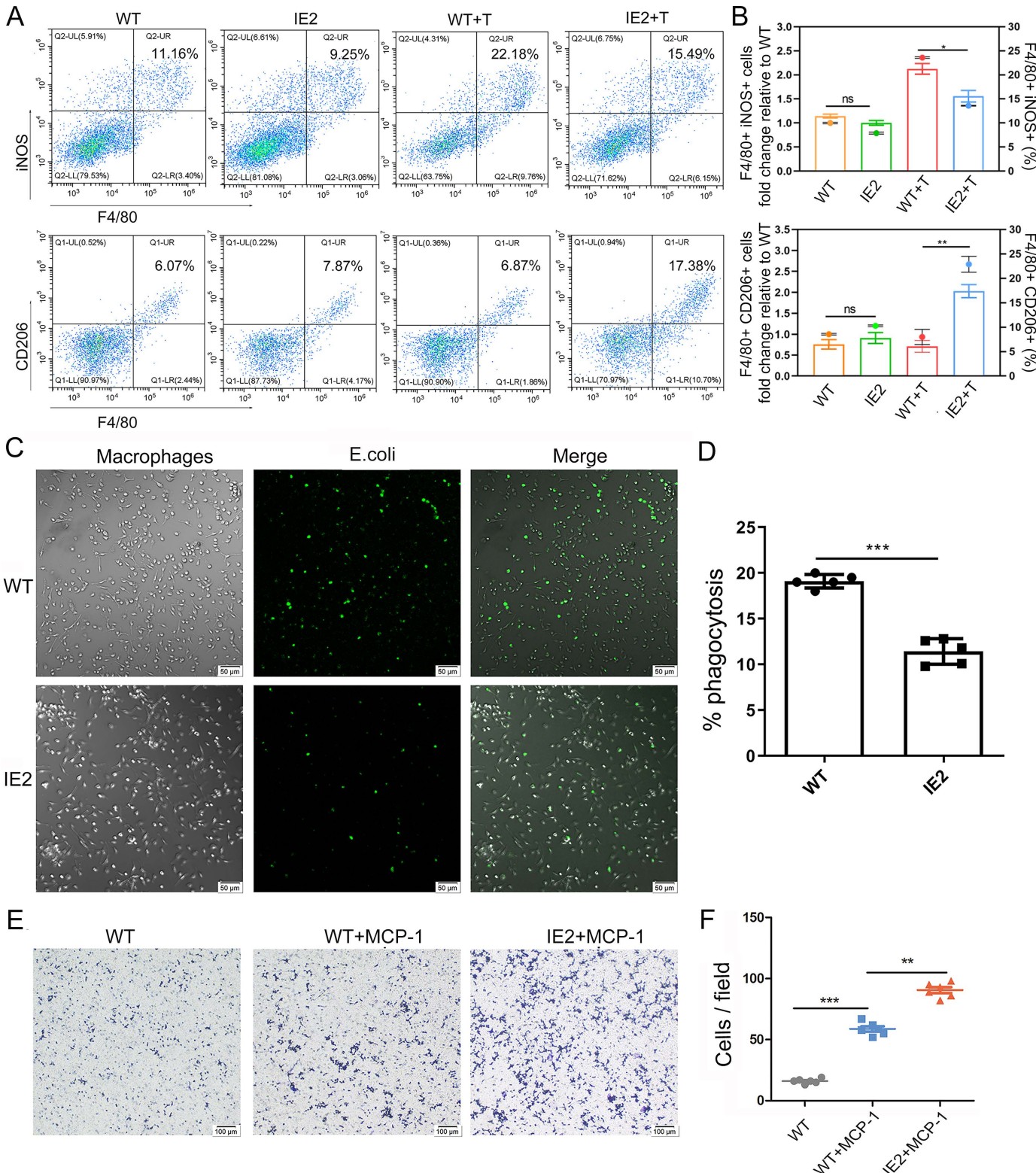

**Fig 3. The effect of IE2 on function of primary macrophages.** (A) M1 phenotype macrophages (F4/80⁺ iNOS⁺) were detected in the spleen by flow cytometry *(right)*. The percentage of M1 phenotype macrophages (F4/80⁺ iNOS⁺) *(left)*. (B) M2 phenotype macrophages (F4/80⁺ CD206⁺) were detected in the spleen by flow cytometry *(right)*. The percentage of M2 phenotype macrophages (F4/80⁺CD206⁺) *(left)*. WT+T means WT mice injected

intraperitoneally with LPS. IE2+T means IE2 mice injected intraperitoneally with LPS. (C) Primary macrophages were obtained and used fluorescence microscope to detect phagocytic function. (D)The percentages of phagocytic in primary macrophages. (E) Primary macrophages recruitment by MCP-1 (20 ng/mL) for 20 min was measured using transwell assay. (F) Summarized data of migration in primary macrophages. Statistically significant differences as determined by t-test from ANOVA. Date are mean±SEM. All experiments were performed independently at least three times, and significance levels were defined as *P < 0.05, **P < 0.01 and ***P < 0.001.

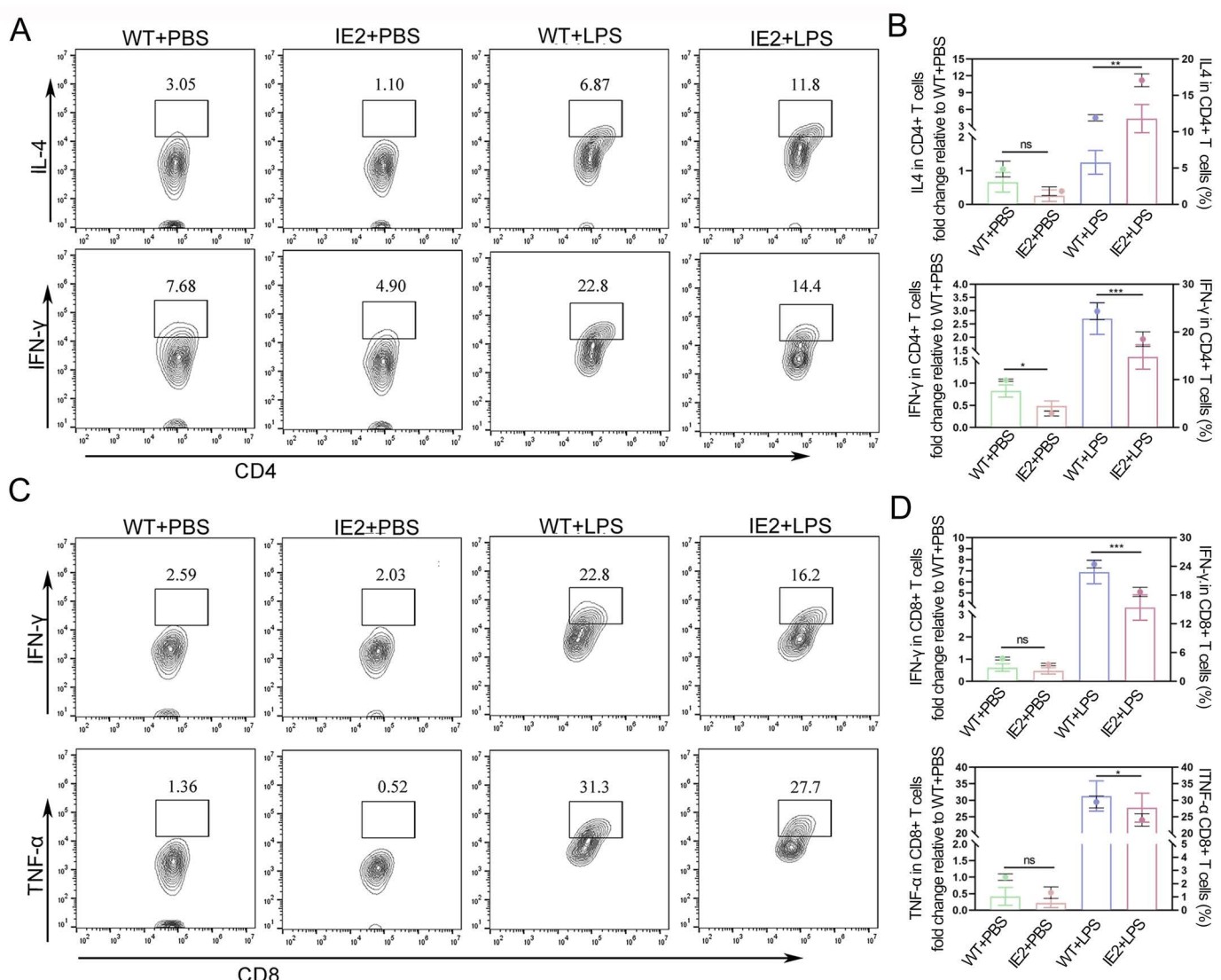

**Fig 4. The effect of IE2 on macrophage stimulated T cell differentiation.** (A) The secretion of IL-4 and IFN-γ in CD4+ T cells were detected by flow cytometry. (B) The percentages of IL-4 and IFN-γ in CD4+ T cells. (C) The secretion of IFN-γ and TNF-α in CD8+ T cells were detected by flow cytometry. (D) The percentages of TNF-α and IFN-γ in CD8+ T cells. Statistically significant differences as determined by t-test from ANOVA. Date are mean±SEM. n = 5. Significance levels were defined as *P < 0.05, **P < 0.01 and ***P < 0.001.

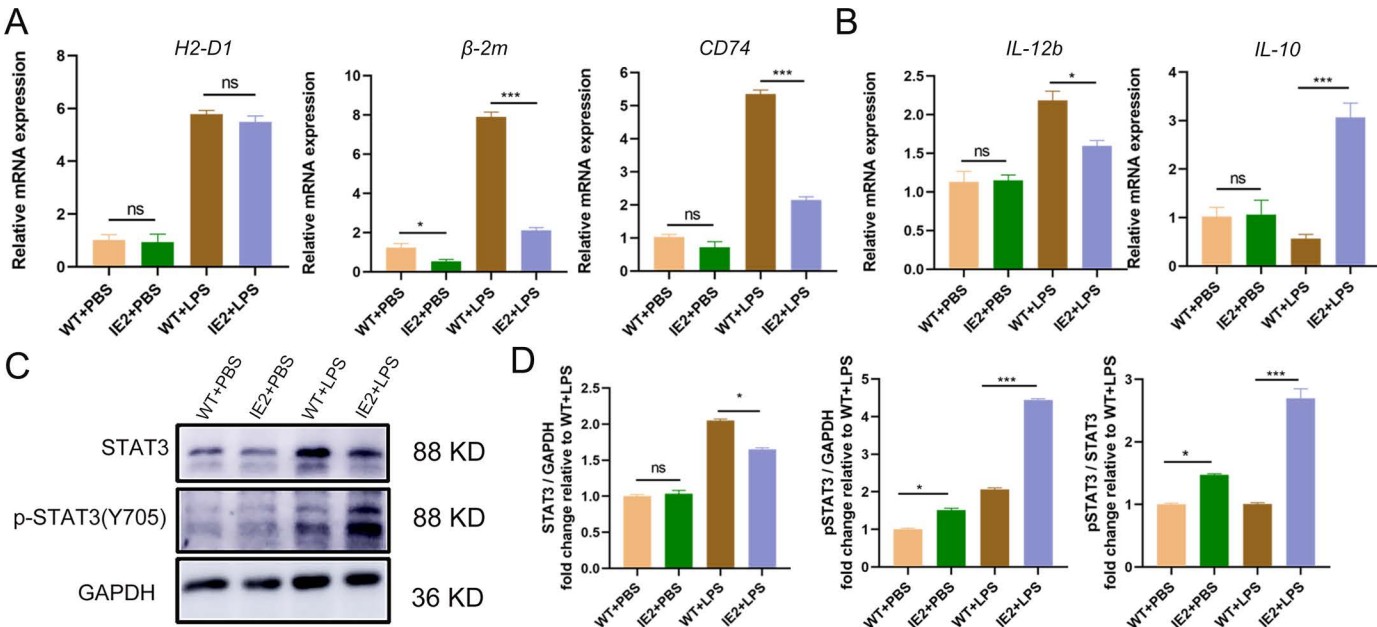

**Fig 5. IE2 up-regulated IL10/STAT3 signaling pathway.** (A) Primary macrophages were cocultured with LPS for 48h. The expression of antigen presentation related molecules mRNA level was detected by RT-qPCR. MHC I antigen presentation pathway: *β-2m* and *H2-D1*; MHC II antigen presentation pathway: *CD74*. (B) The expression of *IL-12b* and *IL-10* mRNA level were detected by RT-qPCR. (C) Whole cell proteins were extracte from the mouse spleen lymphocytes with cultured macrophages, and the STAT3 and p-STAT3 levels were measured was detected by western blot. (D) Summarized data of panel C. Statistically significant differences as determined by t-test from ANOVA. Date are mean±SEM. All experiments were performed independently at least three times, and significance levels were defined as *$P<0.05$ and ***$P<0.001$.

antigen processing through both MHC I and MHC II pathways by capturing antigens to prime T cells. Attenuation of Mφ antigen presentation may be a strategy exploited by HCMV to undermine the host's adaptive immune response to prolong viral persistence and thereby increase the likelihood of horizontal transmission. Previous studies have demonstrated that HCMV has evolved strategies to attenuate Mφ antigen presentation, thereby undermining adaptive immunity and facilitating viral persistence. For instance, HCMV-encoded proteins, including US2, US3, US6, and US11, downregulate MHC I molecules to evade CD8[+] T cell surveillance [21–25]. In permissively infected cells, HCMV gene expression is regulated in a cascade fashion characteristic for herpesviruses. Gene can be categorized into three types: immediate early (IE), early (E), and late (L). HCMV proteins encoded by genes expressed in the immediate-early (IE) phase of infection control the activation of early (E) phase genes [26]. Del Val et al. reported that the E proteins of HCMV prevents the presentation of antigens derived from IE proteins by blocking the transport of peptide-loaded MHC I molecules through the Golgi compartment [27]. In our study, we observed a significant reduction in the number of CD11b[+] Mφs and decreased expression of MHC I and MHC II in IE2 mice. Additionally, IE2 impaired the phagocytic function of Mφs, consistent with previous findings that HCMV-infected monocytes exhibit reduced phagocytic capacity and differentiation [28]. However, the effects of HCMV on Mφ migration remain controversial. While some studies report decreased chemokine-driven migration in HCMV-infected monocytes and Mφs [28,29]. By contrast, Baaschet al. reported that CMV increases the migration ability of bone marrow-derived Mφs *via* the Wnt pathway [30]. Our study showed that IE2 enhances the migratory capacity of Mφs, suggesting that this change in migration ability may promote viral dissemination.

Cell-mediated immunity is critical for controlling HCMV infection and preventing disease progression [31–36]. CD4[+] T cells, activated by MHC II-restricted antigen presentation, play a central role in this process. HCMV-specific CD4[+] T cells control infection through IFN-γ release and MHC II -restricted lysis of infected cells [37–39]. In this study, we found that

IE2 downregulated IFN-γ expression in CD4+ T cells and reduced TNF-α and IFN-γ levels in CD8＋T cells, while increasing IL-4 expression in CD4+ T cells. These findings suggest that IE2 suppresses T cell-mediated immune responses, favoring a Th2-type response over a Th1-type response.

HCMV employs multiple strategies to evade immune detection. For example, HCMV inhibits IFN-γ-stimulated MHC II expression by blocking the JAK/STAT signaling pathway [40]. Park et al. noted that HCMV-encoded US7 and US8 glycoproteins function as innate immune response suppressors by targeting TLR3 and TLR4 [41]. In addition, HCMV US2 degrades components of the MHC class II pathway, such as HLA-DRα and HLA-DMα, preventing recognition by CD4+ T cells [42]. Of note, IE2 mediates the degradation of immune-associated proteins, including STING and CD83 [9,43]. Meanwhile, Botto et al. demonstrated that HCMV-encoded IE86 can block IL-1β protein secretion, inhibiting the innate immune response to escape immune surveillance [44]. Taylor et al. indicated that IE2 could suppress virus-induced pro-inflammatory cytokine mRNA expression, thereby attenuating host innate immune responses [45]. Our data reveal that the IE2 inhibited the immune response by up-regulating the expression of IL-10 and activating the expression and phosphorylation of the downstream signal, STAT3.

IE2 is a multifunctional protein implicated in various biological processes, including the inhibition of neuronal development, promotion of gliomagenesis, and contribution to atherosclerosis [46–49]. However, in this study, our experiments were conducted using mice that were only six-eight weeks old. And at this early stage, we did not observe significant changes in the body weight or dietary intake of the transgenic mice, suggesting that the physiological state of these mice remained relatively stable. This stability allowed us to isolate the immune-modulatory effects of IE2 without confounding factors.

This study has several limitations. First, the complexity of the HCMV genome means that our findings do not fully capture the mechanisms of HCMV immune evasion. Second, there is a lack of studies on the expression levels of IE2 in humans following HCMV infection, and thus our findings are largely speculative about the potential effects of IE2 on macrophages and the immune response in vivo. Additionally, as mice age, additional factors may come into play that could influence immune responses and overall health. Thus, while our model provides valuable insights into the early immune effects of IE2, it may not fully capture the complexities of its impact over the lifespan of the mice. To address these limitations, we recommend that future research explore immune responses in mice at various age stages. This approach will help to elucidate the biological effects of IE2 more comprehensively and provide a clearer understanding of its implications in immunological studies.

In summary, our study suggests a mechanism underlying the IE2-mediated immunosuppressive effects. In IE2 mice, long-term stable expression of IE2 impairs Mφs function by upregulating IL-10 and enhancing the activation of the downstream mediator STAT3, ultimately inhibiting T cell immune responses. Our data provide novel insights into how HCMV evades immune surveillance and establishes long-term persistence in the host.

## Supporting information

**S1 Fig. Construction C57BL/6-Tg (HCMV-*UL122*) transgenic mice (IE2 mice) model flow chart.**
(TIF)

**S2 Fig. Gating strategy.** (A) Innate immune cell subtype gating strategies. (B) The gating strategy was used to identify CD3+ T cells, CD8+ T cells, CD4+ T cells, and their secreted cytokines.
(TIF)

**S3 Fig. Effects of IE2 on body weight and diet in mice.** (A) Body weight changes in different groups. (B) Food intake changes in different groups.
(TIF)

**S4 Raw Images.** The raw images of western blot.
(TIF)

**S5 Raw Data.** All raw data of statistical chart in figures.
(XLSX)

## Author contributions

**Data curation:** Guanghui Song.

**Formal analysis:** Xianjuan Zhang .

**Funding acquisition:** Bin Wang, Yunyang Wang.

**Investigation:** Qing Wang.

**Methodology:** Xianjuan Zhang , Shuo Han.

**Project administration:** Guanghui Song, Bin Wang, Yunyang Wang.

**Resources:** Xianjuan Zhang .

**Software:** Xianjuan Zhang , Shuo Han.

**Supervision:** Qing Wang, Yunyang Wang.

**Writing – original draft:** Xianjuan Zhang .

**Writing – review & editing:** Xianjuan Zhang , Shuo Han.

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
