## [Decision Letter · Decision Letter 0]

9 Jan 2025

PONE-D-24-54197Human cytomegalovirus-IE2 suppresses antigen presentation of macrophage through the IL10/STAT3 signalling pathway in transgenic mousePLOS ONE

Dear Dr. Xian juan,

Thank you for submitting your manuscript to PLOS ONE. After careful consideration, we feel that it has merit but does not fully meet PLOS ONE’s publication criteria as it currently stands. Therefore, we invite you to submit a revised version of the manuscript that addresses the points raised during the review process.

We look forward to receiving your revised manuscript.

Kind regards,

Chunmei Cai

Academic Editor

PLOS ONE

**Journal Requirements:**

2. To comply with PLOS ONE submissions requirements, in your Methods section, please provide additional information regarding the experiments involving animals and ensure you have included details on (a) methods of sacrifice, (b) methods of anesthesia and/or analgesia, and (c) efforts to alleviate suffering.

This research was funded by Shandong Provincial Science and technology Foundation (grant no. 2019JZZY011009), Qingdao Municipal Science and technology Foundation (grant no. 20-2-3-4-nsh), National Key Research and Development Program of China (grant no.2018YFA0900802), Shandong Provincial Natural Science Foundation (grant no. ZR2021QH254).

Reviewers' comments:

Reviewer's Responses to Questions

**Comments to the Author**

1. Is the manuscript technically sound, and do the data support the conclusions?

Reviewer #1: Partly

Reviewer #2: Partly

2. Has the statistical analysis been performed appropriately and rigorously? 

Reviewer #1: Yes

Reviewer #2: Yes

3. Have the authors made all data underlying the findings in their manuscript fully available?

Reviewer #1: No

Reviewer #2: Yes

4. Is the manuscript presented in an intelligible fashion and written in standard English?

Reviewer #1: No

Reviewer #2: Yes

5. Review Comments to the Author

**Reviewer #1: ** In this study, Dr. Zhang and collaborators take advantage of a transgenic mouse model stably expressing the HCMV IE2 protein to prove that the long-term IE2 expression has a major negative impact on immune activation. Although some of the presented results are suggestive, the primary issue I can envisage is the lack of supporting experiments using human cells. These should be easily obtained with primary human macrophages transduced with a plasmid carrying the IE2 gene and with HCMV strains deleted for this gene. The author should explain why they did not pursue these approaches.

In general, the findings are poorly explained, and the majority of the experimental details are vague. Finally, the English language need accurate revision.

Specific details:

- the construct for transgene generation is poorly described;

- in Fig 2 and Fig 3: the authors should mention in the text or at least in the legend what T stands for (in WT+T and IE2+T)

- for Fig 3: a few details are provided in both the text and the legends. Please amend

- Fig 3C : please change “E Coil” to “E. coli”

- Fig 3E,F: how much MCP-1 was used for the transwell experiment? Please include details in the main text and/or in the M&M section

- In numerous places throughout the text, the MCP-1 acronym is wrongly reported as Mcp-1.

- M&M and legends are poorly detailed, with some inaccuracies (i.e., line 197: The Transwell® system WERE used)

**Reviewer #2:**  Summary:

The authors have developed a new transgenic mouse model that allows for investigation of the consequences of prolonged expression of the HCMV gene IE2 on immune activation in mice. Using this model, they found that prolonged expression of IE2 alters macrophage polarization and motility, the consequences of antigen-presenting to CD4 and CD8 T-cells, and that this might be due to IE2 expression leading to higher levels of IL-10 and STAT3 phosphorylation. The new model is interesting and could be potentially useful to others interested in studying functions of IE2. The data generally support the conclusions posited by the authors. The biggest concern I have is regarding the lack of discussion/presentation of any data regarding any other effects on the mice with prolonged expression of IE2. IE2 has many reported functions, not just immunomodulation, that could alter the biology of the mouse in important ways. Other minor concerns are enumerated below. Recommendation: minor revisions, focus on adding in more information about the mice.

Specific areas for improvement:

Major issues:

1. This is a new transgenic mouse model. While Figure 1 adequately shows where in the genome IE2 was knocked-in and identified mice with the knock-in by PCR, it’s important to provide more general information about the health of the mice – do they live as long as WT counterparts? Do they have any unexpected abnormalities? I am sure the authors have noted these things in the development of the transgenic mice – as a reader, I want to know these things to aid in my interpretation of the rest of the paper.

2. Line 52, p.3: “…mouse model simulates the expression process of viral proteins after HCMV infection in vivo to a certain extent.” – there is no data provided for this…? To WHAT extent? It is important to show more characterization of this mouse model, especially when claiming that it recapitulates the expression process. HCMV expresses genes in the typical cascade of IE->E->L, and this model relies on long-term sustained expression of IE2. How is this simulating the “normal” expression of IE2? Data on this would be extremely useful to aid readers in interpreting the reliability of extrapolating data from this study to what we think is going in in humans/human cells.

Minor issues:

1. Line 40 p.2: “macrophages are the first targets during HCMV infection.” – it is often said that epithelial, endothelial, or fibroblast cells are the first cells infected when a host encounters the virus, and that trafficking from these outer tissues to the endothelium allows the virus to encounter cells like macrophages. I definitely agree they are important for viral persistence and spread, but I would not definitively declare macrophages as the first target cells of HCMV.

2. Entire introduction – I would like more specific information included in the introduction, describing a bit more specifically what the cited studies showed. For example, line 43 “Previous studies have shown that HCMV infection mediates immune escape by regulating macrophage activity.” – how does it regulate it? What was shown? What part of macrophage activity was regulated? Was it IE2 or something else? What kinetic class of viral gene (if known) was implicated in regulating macrophage activity? What models were used? Etc

3. Fig. 1A – This is a nice map of the insertion, but it could use some more detail. Under what promoter was IE2 expression placed? The font is also quite small – readability could be improved by altering font sizes.

4. Lines 148-149: “indicating that the UL122 gene insertion does not affect the expression of neighboring genes.” – The assumption is the gene was inserted far enough away from neighboring genes to not alter expression. Including control RTqPCR or Western blots for Fam47c and Cfap47-205 in WT vs IE2 mice would strengthen this argument.

5. Line 158: “In the spleen, Mos were stained with CD11b+” – were cells isolated from the spleen specifically and then stained? Or is this some in vivo staining?

6. Throughout (example line 171) – specify that it is continued/sustained IE2 expression resulting in your phenotypes.

7. Figure 2B: the data are compelling on their own, but my first thought was to calculate in my head the fold changes relative to untreated for each mouse. It would be helpful to include in the figure the same data plotted relative to untreated (as a fold change)

8. Figure 3A-B: as a non-immunologist, it took awhile to parse out which data meant M1 vs M2. This could easily be fixed by adding labels to the figure, and strengthen the figure by making it more readable to people outside the field.

9. Lines 185-188: The experiment could use more explanation (again mainly for those readers who don’t frequently do phagocytosis assays) – where does the GFP come from? What do the images tell us specifically?

10. Lines 189-192: this simple inclusion of “the chemokine MCP-1 is known to induce migration” is a great example of the minor addition of explanation (needed for the phagocytosis assay) really making a difference for my understanding. This was great!

11. Fig 3E legend (line 197) – this could use more detail.

12. Effect of IE2 on T cells (lines 201-213): This section could use a bit more detailed explanation of the question you’re asking, what you measured by flow cytometry, and what it means for both CD4 helper T cells AND CD8 cytotoxic T cells (line 213 seems specifically focused on CD4 – what about CD8? What are the implications there?)

13. Figure 4: As mentioned for figure 2B (point 7 above), it would be useful to plot fold changes relative to the PBS control for each mouse in addition to the data presented. This would clearly show if there’s a specific change in activation as opposed to baseline changes overall in the IE2 mouse (this would also be interesting on its own!)

14. Figure 5: Again, this looks at the changes in IE2 vs WT in LPS-stimulated cells; however, it would be interesting to also include expression of these MHC markers or cytokines at baseline (without LPS) for IE2 and WT mice. The fold change compared to baseline would again be helpful. Same for the STAT3 Western blots.

15. Line 263: what is “E” gene product?

16. Line 293: This study revealed a correlation between IE2 expression and changes in IL-10/STAT3, but the mechanism feels lacking in data. This conclusion would be more fully supported if trans-complementing studies or something could be done – I am not sure whether that is meant to be in the scope of this paper or not. Thus, you could simply say “our study SUGGESTS a mechanism” instead of “our study REVEALS”

17. The last sentence regarding vaccine development is not necessary.

Final thoughts: these data suggest quite a cool link between IE2, STAT3/IL10, and macrophage activity. Looking forward to seeing more in depth follow up studies in the future.

6. PLOS authors have the option to publish the peer review history of their article (what does this mean? ). If published, this will include your full peer review and any attached files.

**Do you want your identity to be public for this peer review?** For information about this choice, including consent withdrawal, please see our Privacy Policy .

Reviewer #1: No

Reviewer #2: No

---

## [Author Response · Author response to Decision Letter 0]

13 Feb 2025

Dear Editors and Reviewers,

Thanks very much for taking your time to review this manuscript (manuscript number: PONE-D-24-54197R1). I really appreciate all your comments and suggestions! These suggestions have enabled us to improve our work. Based on the instructions provided in your letter, we revised the original and marked the modified part by red in revised manuscript (with changes marked). In addition, in order to make the expression of the manuscript more standardized, we have carried out language editing, which may lead to the unmarked parts in the text are not completely consistent with the original manuscript, but the content is the same. Please find my itemized the point to point responses in below. In case of further queries, we are happy to clarify any details and look forward to your reply.

Thanks again!

Sincerely,

Xianjuan Zhang

Corresponding author: Bin WangYunyang Wang

E-mail: wangbin532@126.comwangyy_09@outlook.com.

Journal Requirements:

1.Thank you for stating the following financial disclosure:

This research was funded by Shandong Provincial Science and technology Foundation (grant no. 2019JZZY011009), Qingdao Municipal Science and technology Foundation (grant no. 20-2-3-4- nsh), National Key Research and Development Program of China (grant no.2018YFA0900802), Shandong Provincial Natural Science Foundation (grant no. ZR2021QH254).

Response: Thank you for your attention to our financial disclosure statement. As requested, we hereby confirm the role of the funders in the study as follows: The funders had no role in study design, data collection and analysis, decision to publish, or preparation of the manuscript. We will include this statement in our cover letter for submission.

Response: Thank you for your feedback and for bringing PLOS ONE’s updated requirements for blot and gel reporting to our attention. We have carefully reviewed the journal’s guidelines at the provided links and ensured that our figures and supporting materials fully comply with these requirements. We have prepared the original uncropped and unadjusted images underlying all blot and gel results reported in the Supporting Information files accompanying our revised submission. In our cover letter, we have included the following statement to confirm compliance with PLOS ONE’s requirements: The original uncropped and unadjusted images underlying all blot and gel results reported in this manuscript are provided in the Supporting Information.

3. "We note that your Data Availability Statement is currently as follows: [All relevant data are within the manuscript and its Supporting Information files.]

If there are ethical or legal restrictions on sharing a deidentified data set, please explain them in detail (e.g., data contain potentially sensitive information, data are owned by a thirdparty organization, etc.) and who has imposed them (e.g., an ethics committee). Please also provide contact information for a data access committee, ethics committee, or other institutional body to which data requests may be sent. If data are owned by a third party, please indicate how others may request data access."

Response: Thank you for your note regarding our Data Availability Statement. We confirm that our submission includes the minimal data set required to replicate the results of our study, as defined by PLOS ONE. All raw data, including the values behind means, standard deviations, and other measures, the data used to build graphs, and the points extracted from images for analysis, are provided within the Supporting Information files accompanying our manuscript.

4. “Thank you for responding to our previous query.

please provide additional information regarding the experiments involving animals and ensure you have included details on methods of sacrifice. We hope you understand the reasons behind this request and look forward to hearing from you."

Thank you for your follow-up query regarding the experiments involving animals. We appreciate the importance of providing detailed information to ensure transparency and reproducibility in our research.

Response: In response to your request, we have updated the Materials and Methods

Section (lines 58-68) of our manuscript to include additional details regarding the animal experiments, specifically the methods of sacrifice.

Comments to the Author:

Reviewer #1:

In this study, Dr. Zhang and collaborators take advantage of a transgenic mouse model stably expressing the HCMV IE2 protein to prove that the long-term IE2 expression has a major negative impact on immune activation. Although some of the presented results are suggestive, the primary issue I can envisage is the lack of supporting experiments using human cells. These should be easily obtained with primary human macrophages transduced with a plasmid carrying the IE2 gene and with HCMV strains deleted for this gene. The author should explain why they did not pursue these approaches.

In general, the findings are poorly explained, and the majority of the experimental details are vague. Finally, the English language need accurate revision.

Response: Thank you for your valuable comments. We appreciate the opportunity to address your concerns. Our study focuses on the mechanisms of IE2 in a mouse model, which provides a systematic biological context for understanding its function. Additionally, mouse models are commonly used in immunology, and their macrophages share similarities with human macrophages. Thus, our findings can offer insights into HCMV's role in humans. We intend to investigate IE2's role in primary human macrophages and conduct experiments with HCMV strains lacking the IE2 gene in future studies. We hope this clarifies our approach and addresses your concerns. Thank you again for your feedback, which has helped us improve our manuscript.

Specific details:

1. the construct for transgene generation is poorly described.

Response: Thank you for your valuable feedback regarding the description of the transgenic construct. We appreciate your suggestion and have expanded the M&M section to provide a more detailed account of the construction process (lines 69-73). We believe this additional information will enhance the clarity and comprehensiveness of our study.

2. in Fig 2 and Fig 3: the authors should mention in the text or at least in the legend what T stands for (in WT+T and IE2+T).

Response: Thank you very much for your insightful comments and suggestions regarding our manuscript. We greatly appreciate your attention to detail, which has helped us improve the clarity of our work. In response to your observation about Figures 2 and 3, we have updated the text and figure legends to explicitly define what "T" represents in the contexts of WT+T and IE2+T (lines 190-191). We believe this clarification enhances the reader's understanding of our findings.

3. for Fig 3: a few details are provided in both the text and the legends. Please amend.

Response: Thank you for your thorough review and valuable feedback on our manuscript. Regarding your comments about the details provided in Fig 3, we have carefully revised both the text and the figure legend (lines 205-219).

4. Fig 3C: please change “E Coil” to “E. coli”

Response: Thank you for your careful review and for pointing out the error in Figure 3C. We have corrected “E. coil” to “E. coli” as you suggested. Correction has been made in the revised version.

5. Fig 3E, F: how much MCP-1 was used for the transwell experiment? Please include details in the main text and/or in the M&M section

Response: Thank you for your valuable feedback regarding the transwell experiments presented in Figures 3E and 3F. We appreciate your suggestion to include more details about the concentration of MCP-1 used in these experiments. We have now added the specific concentration of MCP-1 to the M&M section of the manuscript for clarity (lines 113-116). Additionally, we have referenced this information in the main text to ensure that readers can easily find it (lines 190-191).

6. In numerous places throughout the text, the MCP-1 acronym is wrongly reported as Mcp-1.

Response: Thank you for your valuable feedback regarding our manuscript. We appreciate your attention to detail and the time you have taken to review our work. In response to your comment about the abbreviation for MCP-1, we acknowledge that it was incorrectly reported as "Mcp-1" in several places throughout the manuscript. We have carefully reviewed the text and corrected all instances to the proper format "MCP-1" to ensure consistency and accuracy.

7. M&M and legends are poorly detailed, with some inaccuracies (i.e., line 197: The Transwell® system WERE used)

Response: Thank you for your constructive feedback regarding the M&M section, as well as the legends. We have revised the M&M and legends section to provide more comprehensive details and have corrected the inaccuracies you pointed out. Additionally, we have ensured that all other relevant details are clearly presented and accurate.

Reviewer #2

Major issues:

1. This is a new transgenic mouse model. While Figure 1 adequately shows where in the genome IE2 was knocked-in and identified mice with the knock-in by PCR, its important to provide more general information about the health of the mice–do they live as long as WT counterparts? Do they have any unexpected abnormalities? I am sure the authors have noted these things in the development of the transgenic mice – as a reader, I want to know these things to aid in my interpretation of the rest of the paper.

Response: Thank you for your valuable feedback regarding our newly developed transgenic mouse model. We appreciate your inquiry into any unexpected abnormalities observed during the development of these mice, as this information is indeed important for interpreting our findings. In our study, we have observed that the transgenic mice develop learning and memory impairments and metabolic disturbances after 12 weeks of age. However, it is important to note that the mice used in our experiments were 6-8 weeks old, a developmental stage at which we have not observed any significant impacts on weight or dietary intake. Therefore, we believe that the effects of IE2 on immune responses can be accurately assessed without the confounding factors associated with the later-stage abnormalities. We have added a discussion in the revised manuscript to clarify these observations and their implications for our study (lines 322-339).

2. Line 52, p.3: “…mouse model simulates the expression process of viral proteins after HCMV infection in vivo to a certain extent.” – there is no data provided for this…? To WHAT extent? It is important to show more characterization of this mouse model, especially when claiming that it recapitulates the expression process. HCMV expresses genes in the typical cascade of IE->E->L, and this model relies on long-term sustained expression of IE2. How is this simulating the “normal” expression of IE2? Data on this would be extremely useful to aid readers in interpreting the reliability of extrapolating data from this study to what we think is going in in humans/human cells.

Response: Thank you for your thorough review of our manuscript and for your valuable feedback regarding the mouse model. We appreciate your concerns about the need for data supporting the extent to which our model simulates the expression of viral proteins following HCMV infection. In the revised manuscript, we have included a more detailed description of the characteristics of the mouse model, and added the study's limitations in the discussion section. Our study is based on the construction of an IE2 transgenic mouse, which relies on the prolonged expression of IE2 to simulate its effects on macrophages following HCMV infection. While we acknowledge the limitation of not being able to quantitatively measure IE2 expression levels directly, we emphasize the significance of this model in understanding the role of IE2 in modulating immune responses.

Minor issues:

1. Line 40 p.2: “macrophages are the first targets during HCMV infection.” – it is often said that epithelial, endothelial, or fibroblast cells are the first cells infected when a host encounters the virus, and that trafficking from these outer tissues to the endothelium allows the virus to encounter cells like macrophages. I definitely agree they are important for viral persistence and spread, but I would not definitively declare macrophages as the first target cells of HCMV.

Response: Thank you for your insightful comment. We appreciate your perspective on the initial target cells during HCMV infection. While our previous statement that macrophages are the first targets of HCMV infection was based on [1]. We acknowledge that the literature also supports epithelial, endothelial, and fibroblast cells as primary targets. To avoid ambiguity, we have revised the text to clarify that while macrophages play a significant role in the persistence and dissemination of HCMV. We have made this adjustment to ensure a more accurate representation of the infection dynamics in our manuscript (lines 39-40).

1. Bayer C, Varani S, Wang L, Walther P, Zhou S, Straschewski S, Bachem M, Söderberg-Naucler C, Mertens T, Frascaroli G. Human cytomegalovirus infection of M1 and M2 macrophages triggers inflammation and autologous T-cell proliferation. J Virol. 2013 Jan;87(1):67-79. doi: 10.1128/JVI.01585-12. Epub 2012 Oct 10. PMID: 23055571; PMCID: PMC3536399.

2. Entire introduction – I would like more specific information included

---

## [Decision Letter · Decision Letter 1]

10 Mar 2025

PONE-D-24-54197R1Human cytomegalovirus-IE2 suppresses antigen presentation of macrophage through the IL10/STAT3 signalling pathway in transgenic mousePLOS ONE

Dear Dr. Zhang ,

Thank you for submitting your manuscript to PLOS ONE. After careful consideration, we feel that it has merit but does not fully meet PLOS ONE’s publication criteria as it currently stands. Therefore, we invite you to submit a revised version of the manuscript that addresses the points raised during the review process.

We look forward to receiving your revised manuscript.

Kind regards,

Chunmei Cai

Academic Editor

PLOS ONE

Journal Requirements:

Reviewers' comments:

Reviewer's Responses to Questions

**Comments to the Author**

1. If the authors have adequately addressed your comments raised in a previous round of review and you feel that this manuscript is now acceptable for publication, you may indicate that here to bypass the “Comments to the Author” section, enter your conflict of interest statement in the “Confidential to Editor” section, and submit your "Accept" recommendation.

Reviewer #1: (No Response)

Reviewer #2: All comments have been addressed

2. Is the manuscript technically sound, and do the data support the conclusions?

Reviewer #1: Yes

Reviewer #2: Yes

3. Has the statistical analysis been performed appropriately and rigorously? 

Reviewer #1: Yes

Reviewer #2: Yes

4. Have the authors made all data underlying the findings in their manuscript fully available?

Reviewer #1: Yes

Reviewer #2: Yes

5. Is the manuscript presented in an intelligible fashion and written in standard English?

Reviewer #1: No

Reviewer #2: Yes

6. Review Comments to the Author

Reviewer #1: While the authors have adequately addressed the main concerns related to the first round of revisions, the following should still be amended:

- The authors have added some details about the generation of the transgene model, both in the M&M section and in the new Fig. S1. However, in the M&M the new paragraph sounds: “A pAV.Ex1d-CMV-IE2 vector containing a cDNA fragment of the IE2 gene was constructed, and the vector was injected into…”. No details are provided about the origin of the viral gene: where does this gene come from? What strategy did they employ for its amplification and cloning?

- Furthermore, the quality of figure S1 is poor, with some black squares partially covering the image and a scheme from the company's website that I'm not convinced can be used in this context.

- Finally, some more editing of the English language is required before acceptance.

Reviewer #2: The authors have adequately addressed reviewer concerns, which have strengthened the paper to be of higher quality for publication.

7. PLOS authors have the option to publish the peer review history of their article (what does this mean? ). If published, this will include your full peer review and any attached files.

**Do you want your identity to be public for this peer review?** For information about this choice, including consent withdrawal, please see our Privacy Policy .

Reviewer #1: No

Reviewer #2: No

---

## [Author Response · Author response to Decision Letter 1]

16 Mar 2025

Dear Editors and Reviewers,

Thank you very much for taking the time to review our manuscript (Manuscript Number: PONE-D-24-54197R2). We greatly appreciate your insightful comments and constructive suggestions, which have significantly enhanced the quality of our work.

In response to your feedback, we have carefully revised the manuscript accordingly. All modifications have been highlighted in red in the revised version. Additionally, to ensure the manuscript meets high linguistic standards, we have conducted professional language editing. While this may result in some unmarked textual variations from the original manuscript, we assure you that the core content remains unchanged.

Please find below our point-by-point responses to your comments. Should you have any further questions or require additional clarification, we would be more than happy to provide further details. We look forward to hearing from you.

Thanks again!

Sincerely,

Xianjuan Zhang

Corresponding author: Bin WangYunyang Wang

E-mail: wangbin532@126.comwangyy_09@outlook.com.

Comments to the Author:

Reviewer #1:

1. The authors have added some details about the generation of the transgene model, both in the M&M section and in the new Fig. S1. However, in the M&M the new paragraph sounds: “A pAV.Ex1d-CMV-IE2 vector containing a cDNA fragment of the IE2 gene was constructed, and the vector was injected into…”. No details are provided about the origin of the viral gene: where does this gene come from? What strategy did they employ for its amplification and cloning?

Response: Thank you very much for your insightful comments and suggestions regarding our manuscript. We greatly appreciate your attention to detail, which has helped us improve the clarity of our work. In response to your suggestion, we have now provided comprehensive details about the viral gene origin and cloning strategy in the revised M&M section (lines 73-87).

2. Furthermore, the quality of figure S1 is poor, with some black squares partially covering the image and a scheme from the company's website that I'm not convinced can be used in this context.

Response: Thank you for bringing these issues with Figure S1 to our attention. We sincerely apologize for the poor image quality. We have completely revised Figure S1 in the updated manuscript to address these concerns.

3. Finally, some more editing of the English language is required before acceptance.

Response: Thank you for pointing out the need for additional language editing. We sincerely appreciate your careful reading of our manuscript and your valuable suggestion. In order to improve the English language quality throughout the manuscript, the manuscript has been professionally edited by Elsevier Language Editing Services (Order reference: ASLESTD1076952). We believe these professional edits have significantly improved the clarity and readability of our manuscript while maintaining its scientific integrity.

Reviewer #2

The authors have adequately addressed reviewer concerns, which have strengthened the paper to be of higher quality for publication.

Response: We are sincerely grateful for your positive feedback and for recognizing our efforts. We greatly appreciate the time and expertise you have dedicated to improving our manuscript. We are honored by your assessment that our manuscript now meets the journal's publication standards. Thank you again for your valuable guidance, which has undoubtedly strengthened our work.

---

## [Decision Letter · Decision Letter 2]

21 Mar 2025

Human cytomegalovirus-IE2 suppresses antigen presentation of macrophage through the IL10/STAT3 signalling pathway in transgenic mouse

PONE-D-24-54197R2

Dear Dr. Xianjuan Zhang,

We’re pleased to inform you that your manuscript has been judged scientifically suitable for publication and will be formally accepted for publication once it meets all outstanding technical requirements.

Kind regards,

Chunmei Cai

Academic Editor

PLOS ONE

Additional Editor Comments (optional):

Reviewers' comments:

Reviewer's Responses to Questions

**Comments to the Author**

1. If the authors have adequately addressed your comments raised in a previous round of review and you feel that this manuscript is now acceptable for publication, you may indicate that here to bypass the “Comments to the Author” section, enter your conflict of interest statement in the “Confidential to Editor” section, and submit your "Accept" recommendation.

Reviewer #1: All comments have been addressed

2. Is the manuscript technically sound, and do the data support the conclusions?

Reviewer #1: Yes

3. Has the statistical analysis been performed appropriately and rigorously? 

Reviewer #1: Yes

4. Have the authors made all data underlying the findings in their manuscript fully available?

Reviewer #1: Yes

5. Is the manuscript presented in an intelligible fashion and written in standard English?

Reviewer #1: Yes

6. Review Comments to the Author

Reviewer #1: (No Response)

7. PLOS authors have the option to publish the peer review history of their article (what does this mean? ). If published, this will include your full peer review and any attached files.

**Do you want your identity to be public for this peer review?** For information about this choice, including consent withdrawal, please see our Privacy Policy .

Reviewer #1: No

---

## [Editor Report · Acceptance letter]

PONE-D-24-54197R2

PLOS ONE

Dear Dr. Zhang ,

I'm pleased to inform you that your manuscript has been deemed suitable for publication in PLOS ONE. Congratulations! Your manuscript is now being handed over to our production team.

Kind regards,

on behalf of

Dr. Chunmei Cai

Academic Editor

PLOS ONE